# Determining propensity for sub-optimal low-density lipoprotein cholesterol response to statins and future risk of cardiovascular disease

Stephen Franklin Weng[1,☯], Ralph Kwame Akyea[1,☯]*, Kenneth KC Man[2,3], Wallis C. Y. Lau[2,3], Barbara Iyen[1], Joseph Edgar Blais[3], Esther W. Chan[3], Chung Wah Siu[4], Nadeem Qureshi[1‡], Ian C. K. Wong[2,3‡], Joe Kai[1‡]

**1** Primary Care Stratified Medicine (PRISM), Division of Primary Care, University of Nottingham, Nottingham, United Kingdom, **2** Centre for Medicine Optimisation Research and Education (CMORE), Research Department of Practice and Policy, UCL School of Pharmacy, London, United Kingdom, **3** Centre for Safe Medication Practice and Research, Department of Pharmacology and Pharmacy, Li Ka Shing Faculty of Medicine, University of Hong Kong, Hong Kong, China, **4** Cardiology Division, Department of Medicine, Li Ka Shing Faculty of Medicine, University of Hong Kong, Hong Kong, China

☯ These authors contributed equally to this work.
‡ These authors also contributed equally to this work.
* Ralph.Akyea1@nottingham.ac.uk

**Data Availability Statement:** The third-party data for this study were obtained from the Clinical Practice Research Datalink (CPRD) and the University of Hong Kong/Hospital Authority Hong

## Abstract

### Background

Variability in low-density lipoprotein cholesterol (LDL-C) response to statins is underappreciated. We characterised patients by their statin response (SR), baseline risk of cardiovascular disease (CVD) and 10-year CVD outcomes.

### Methods and results

A multivariable model was developed using 183,213 United Kingdom (UK) patients without CVD to predict probability of sub-optimal SR, defined by guidelines as <40% reduction in LDL-C. We externally validated the model in a Hong Kong (HK) cohort (n = 170,904). Patients were stratified into four groups by predicted SR and 10-year CVD risk score: [SR1] optimal SR & low risk; [SR2] sub-optimal SR & low risk; [SR3] optimal SR & high risk; [SR4] sub-optimal SR & high risk; and 10-year hazard ratios (HR) determined for first major adverse cardiovascular event (MACE). Our SR model included 12 characteristics, with an area under the curve of 0.70 (95% confidence interval [CI] 0.70–0.71; UK) and 0.68 (95% CI 0.67–0.68; HK). HRs for MACE in predicted sub-optimal SR with low CVD risk groups (SR2 to SR1) were 1.39 (95% CI 1.35–1.43, p<0.001; UK) and 1.14 (95% CI 1.11–1.17, p<0.001; HK). In both cohorts, patients with predicted sub-optimal SR with high CVD risk (SR4 to SR3) had elevated risk of MACE (UK HR 1.36, 95% CI 1.32–1.40, p<0.001: HK HR 1.25, 95% CI 1.21–1.28, p<0.001).

Kong West Cluster. CPRD is a research service that provides primary care and linked data for public health research. CPRD and University of Hong Kong/Hospital Authority data governance and our own license to use data do not allow us to distribute or make available patient data directly to other parties. However, data is available upon application to CPRD (www.cprd.com) and the University of Hong Kong/Hospital Authority (www. ha.org.hk).

**Funding:** Funded by scientific donation by AMGEN Ltd. The funders of the study had no role in study design, data collection, data analysis, data interpretation, or writing of the report.

**Competing interests:** NQ is a member of the most recent NICE Familial Hypercholesterolaemia & Lipid Modification Guideline Development Groups (CG71 & CG181). SW, IW are members of the Clinical Practice Research Datalink (CPRD) Independent Scientific Advisory Committee (ISAC). RKA currently holds an NIHR-SPCR funded studentship (2018-2021). SW reports honorarium from AMGEN and is also an employee of Janssen. KM holds the CW Malpethorpe Fellowship Award and reports personal fees from IQVIA Ltd have been received outside the submitted work. JB is supported by the Hong Kong Research Grants Council as a recipient of the Hong Kong PhD Fellowship Scheme. EC and IW report grants from Amgen. The remaining authors have no competing interests. This does not alter our adherence to PLOS ONE policies on sharing data and materials.

## Conclusions

Patients with sub-optimal response to statins experienced significantly more MACE, regardless of baseline CVD risk. To enhance cholesterol management for primary prevention, statin response should be considered alongside risk assessment.

## Introduction

Cardiovascular disease (CVD) is a significant cause of mortality and morbidity, accounting for almost a third of all deaths globally [1]. Statins, the most widely prescribed class of drug worldwide, are an effective and low-cost solution in lowering cholesterol levels and reducing future risk of CVD [2]. Statin use is set to rise further in many countries as guidelines recommend their use in greater proportions of the population [3, 4]. In the United States alone, the proportion of adults whom statins are indicated has increased from 17.9% (21.8 million) to 27.8% (39.2 million) from 2002 to 2013 [5], with similar magnitude increases in most European [6] and Asian countries [7].

Following meta-analyses of cholesterol treatment trials [8, 9], the safety and intended percentage reductions in LDL-C achieved by specific statins (and their doses) have been well-established. Guidelines in the United States (US), United Kingdom (UK), Europe (EU) and Hong Kong (HK) recommend intended LDL-C reduction targets for statin therapy to reduce CVD. The 2013 American College of Cardiology/American Heart Association (ACC/AHA) guideline suggests a fixed dose (or intensity) of statin depending on absolute CVD risk, with intended LDL-C reduction of 30–50% [4]. The UK National Institute of Clinical Excellence (NICE) Guideline aims for a greater than 40% reduction in non-high-density lipoprotein cholesterol (non-HDL-C) [3]. An LDL-C reduction of 50% is recommended by European Society of Cardiology guidelines [10]. Hong Kong primary prevention guidelines follow similar ACC/AHA, NICE, and ESC guideline targets for LDL-C reduction [11].

These guidelines are designed to curb CVD deaths; however, recent evidence has shown that over 50% of a cohort of 165,411 patients prescribed statins for primary prevention in the UK fail to reach treatment targets, which significantly increases risk of heart disease and stroke [12]. The reasons are varied, but there is evidence to suggest individual variability in LDL-C response [13] and significant non-adherence [14]. It is unclear which patient characteristics may be associated with variation in LDL-C reduction after commencing statin therapy for primary prevention of CVD and if these clinical characteristics can be used to predict an individual's likelihood of achieving recommended LDL-C targets. There are currently no management strategies in clinical practice to determine if patients will experience sub-optimal reduction in LDL-C when prescribed statins, nor the impact this may have on future CVD outcomes. First, we conducted an international study using two large geographically distinct cohorts to characterise LDL-C response to statin therapy in relation to baseline cardiovascular risk. We then determined the association between LDL-C response, baseline CVD risk and future CVD outcomes.

## Materials and methods

### Data source

This study used the anonymized electronic health records from the UK Clinical Practice Research Datalink (CPRD) and the Hong Kong (HK) Clinical Data Analysis and Reporting System (CDARS).

CPRD contains anonymized patient data for more than 15 million patients and is representative of the general UK population [15]. CPRD is one of the largest quality-assured databases of longitudinal medical records from primary care in the world, with over 800 UK primary medical care practices. Regulatory approval for this study (protocol number: 17_200RA) was obtained from the CPRD Independent Scientific Advisory Committee.

The CDARS contains data from the Hong Kong Hospital Authority, which serves a population of over 7.4 million through 43 hospitals and institutions, 48 specialist outpatient clinics, 73 general outpatient clinics, covering approximately 80% of all hospital admissions in Hong Kong [16]. The use of CDARS data for this study was approved by the Institutional Review Board of the University of Hong Kong/Hospital Authority Hong Kong West Cluster (protocol number: UW 17–135).

## Study population

The study cohorts comprised 183,213 patients initiating statin therapy with repeat prescriptions between 3rd September 1990 and 7th June 2016 in UK CPRD and 170,904 patients between 1st of January 2003 and 31st of December 2011 in HK CDARS. Patients were eligible for inclusion if they were registered with their health care provider for at least 12 months. Patients were required to have at least two recorded LDL-C measurements (directly measured). The baseline LDL measure was within 12 months before or at the initiation of statins and the follow-up measure was within 24 months after initiation of statins. Patients on statin treatment for primary prevention of CVD who failed to achieve greater than 40% reduction in their untreated baseline LDL-C, recorded within the first 24 months, were defined as sub-optimal LDL-C responders. This was based on the lower bound expected reduction of LDL-C recommended in CVD prevention guidelines [3, 10, 11, 17] and also recognised in meta-analyses [8]. Percentage LDL-C reduction, which considers individual baseline levels, has been shown to have better prognostic value than attained LDL-C for future CVD events [18]. For on-treatment response, the most recent LDL-C record at or closest to 24 months was used. This was because many routine annual medication reviews or cholesterol assessments in clinical practice are unlikely to be documented in the EHRs until beyond 12 months.

All individuals with a diagnosis of CVD (defined as coronary heart disease, cerebrovascular disease and peripheral arterial disease) before initiating statin therapy were excluded. Candidate predictors (documented in **S1 Table**) were assessed for inclusion by clinical review and prior evidence.

## Outcomes

The primary outcome for this analysis was the incidence of first major adverse cardiovascular event (MACE) over 10-year follow-up after initiation of statins. We defined MACE as a composite of coronary heart disease (CHD), stroke, transient ischemic attack (TIA), peripheral vascular disease (PVD), and cardiovascular (CV) death [19]. CHD included acute myocardial infarction, coronary revascularisations, stable and unstable angina. CVD events were identified from linked primary, secondary, and death registries. These are coded diagnoses based on National Health Service (NHS) Read codes and ICD-10 in the UK and mapped to ICD-9-CM in HK. Secondary outcomes include all-cause mortality, and CVD outcomes by sub-type.

## Study design and statistical analyses

The study involved two phases, detailed in the study flowchart (**Fig 1**). Phase I involved the development and validation of the statin response (SR) propensity model and Phase II stratified groups according to statin response and evaluated future CVD and mortality outcomes.

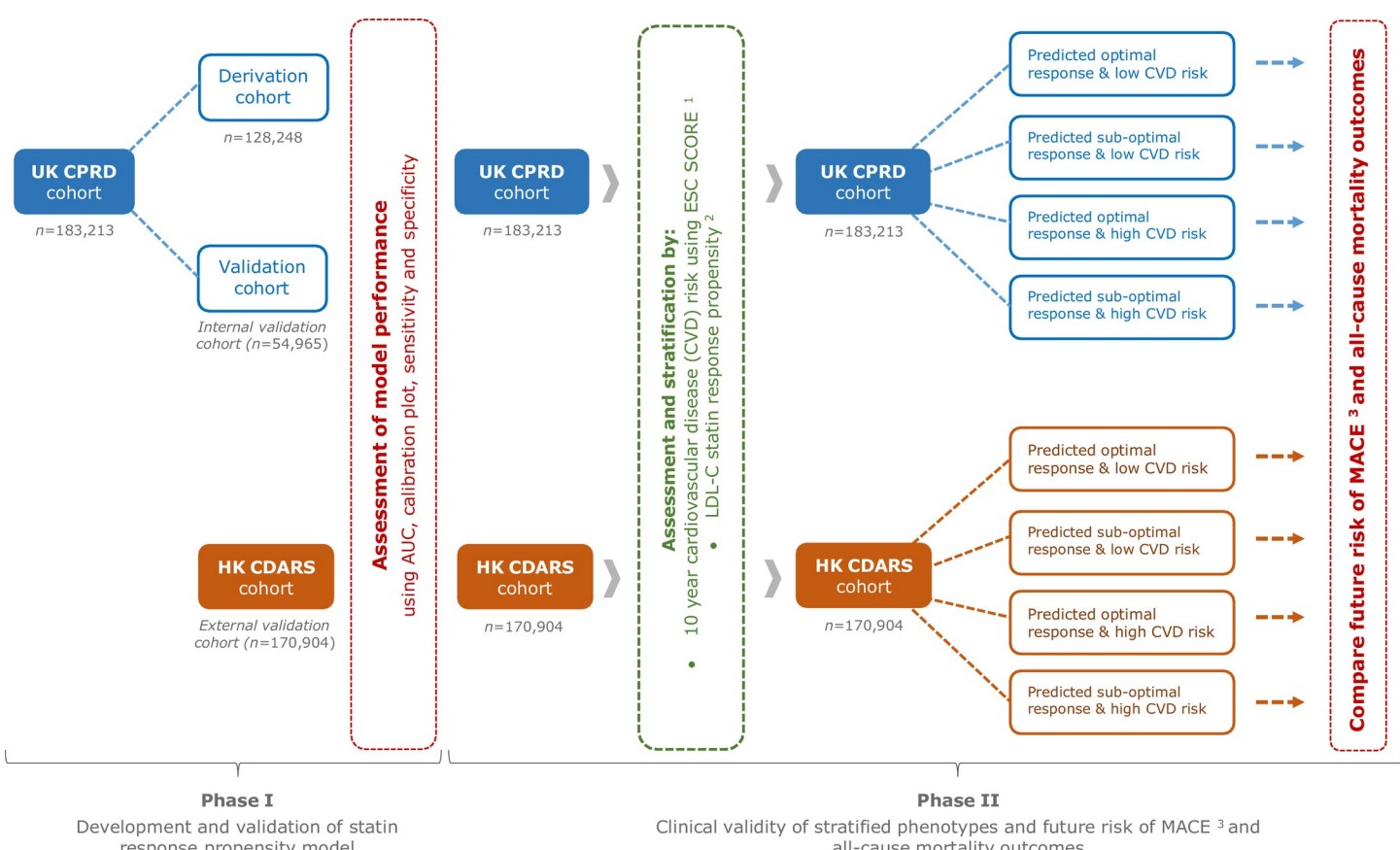

**Figure X. Study Flowchart**

[1] ESC: European Society of Cardiology CVD SCORE >5% is defined as high CVD risk; [2] Low Density Lipoprotein Cholesterol statin response propensity of > 50% is defined as sub-optimal response; [3] MACE – major adverse cardiovascular events is a composite of coronary heart disease, stroke, transient ischemic attack, peripheral vascular disease, and cardiovascular death

**Fig 1.**

Full methods on model development, validation, statistical methods and evaluation of future CVD outcomes are detailed in **S1 Text**. Stata SE version 15 and RStudio were used for all analysis.

## Results

### Characteristics of the cohorts

Within 24 months following initiation of statin therapy, 51.3% of UK patients had sub-optimal cholesterol response in the derivation cohort. However, a higher proportion of HK patients had a sub-optimal LDL-C response (60.8%) compared to the UK. Both derivation and validation cohorts had similar mean age, mean baseline LDL-C, and proportion of females. Baseline mean LDL-C was higher in UK patients than in HK patients (156.5 mg/dL and 156.6 mg/dL for UK cohorts compared to 149.8 mg/dL for the HK cohort). Higher potency statins tended to be prescribed more frequently in in the UK than in HK (with 23.6% prescribed low potency in the UK compared to 68.5% prescribed low potency in HK). Full baseline characteristics are presented in **Table 1**, with **S2 Table** detailing the grouping of statins based on potency.

Disease outcomes over 10-year follow-up for both cohorts are shown in **Fig 2**. Overall, there were 40,928 cases of MACE (22.3%) in the UK population and 37,777 cases of MACE

**Table 1. Clinical characteristics of patients initiating statin treatment and free from cardiovascular disease at baseline (n = 183,213).**

| Characteristic | Units | UK CPRD derivation cohort | UK CPRD internal validation cohort | HK CDARS external validation cohort |
|---|---|---|---|---|
| Total sample size | N | 128,248 | 54,965 | 170,904 |
| Follow-up time (years) | Median (IQR) | 5.29 (2.28–8.78) | 5.26 (2.29–8.70) | 6.92 (3.78–8.61) |
| Sex (Females) | N (%) | 60,891 (47.5) | 26,161 (47.6) | 89,315 (52.3) |
| Sub-optimal LDL-C response to statin | N (%) | 65,777 (51.3) | 28,141 (51.2) | 103,899 (60.8) |
| Age (years) | Mean (SD) | 62.9 (11.9) | 62.9 (11.8) | 63.0 (12.0) |
| | Median (IQR) | 63.0 (55.0–71.0) | 63.0 (55.0–71.0) | 62.8 (54.7–72.0) |
| Baseline LDL-C (mg/dL) | Mean (SD) | 156.5 (44.2) | 156.6 (44.2) | 150.0 (41.2) |
| | Median (IQR) | 154.7 (127.6–184.1) | 154.7 (127.6–184.5) | 147.6 (125.0–171.9) |
| Body mass index (kg/m$^2$) | Mean (SD) | 28.72 (4.63) | 28.70 (4.61) | N/A |
| | Median (IQR) | 28.3 (25.9–30.8) | 28.25 (25.9–30.8) | |
| Systolic blood pressure (mmHg) | Mean (SD) | 142 (19) | 142 (19) | N/A |
| | Median (IQR) | 140 (130–150) | 140 (130–150) | |
| Diastolic blood pressure (mmHg) | Mean (SD) | 82 (10) | 82 (10) | N/A |
| | Median (IQR) | 81 (76–88) | 81 (76–88) | |
| Ethnicity | | | | |
| White | N (%) | 11,026 (8.6) | 4,738 (8.6) | * |
| Asian | | 563 (0.4) | 262 (0.5) | |
| Black/African/Caribbean | | 448 (0.3) | 180 (0.3) | |
| Mixed/multiple | | 684 (0.5) | 282 (0.5) | |
| Other | | 380 (0.3) | 163 (0.3) | |
| Unknown | | 115,147 (89.8) | 49,340 (89.8) | |
| **Comorbidities (prior to 1st statin)** | | | | |
| Atrial fibrillation | N (%) | 4,200 (3.3) | 1,821 (3.3) | 3,828 (2.2) |
| Chronic kidney disease | N (%) | 3,574 (2.8) | 1,592 (2.9) | 3,150 (1.8) |
| Diabetes | N (%) | 20,843 (16.3) | 8,928 (16.2) | 32,457 (19.0) |
| Dyslipidaemias | N (%) | 10,262 (8.0) | 4,375 (8.0) | 10,059 (5.9) |
| Family history of cardiovascular disease | N (%) | 13,491 (10.5) | 5,837 (10.6) | N/A |
| Family history of hyperlipidaemia | N (%) | 281 (0.2) | 120 (0.2) | N/A |
| Treated hypertension | N (%) | 33,631 (26.2) | 14,319 (26.1) | 22,670 (13.3) |
| Hypothyroidism | N (%) | 5,340 (4.2) | 2,243 (4.1) | 2,636 (1.5) |
| Liver disease | N (%) | 1,221 (1.0) | 517 (0.9) | 497 (0.3) |
| Migraine | N (%) | 2,766 (2.2) | 1,182 (2.2) | 359 (0.2) |
| Nephrotic syndrome | N (%) | 83 (0.1) | 40 (0.1) | 1,062 (0.6) |
| Rheumatoid arthritis | N (%) | 1,185 (0.9) | 480 (0.9) | 1,377 (0.8) |
| Systemic lupus erythematosus | N (%) | 157 (0.1) | 54 (0.1) | 622 (0.4) |
| **Medications (prescribed within 12 months prior to 1st statin)** | | | | |
| Medication count | Median (IQR) | 4 (1–6) | 4 (1–6) | 8 (5–12) |
| Antipsychotics | N (%) | 5,421 (4.2) | 2,295 (4.2) | 3,279 (1.9) |
| Other lipid lowering medication | N (%) | 535 (0.4) | 223 (0.4) | 10,885 (6.4) |
| Oral corticosteroids | N (%) | 5,390 (4.2) | 2,396 (4.4) | 8,906 (5.2) |

(*Continued*)

**Table 1.** (Continued)

| Characteristic | Units | UK CPRD derivation cohort | UK CPRD internal validation cohort | HK CDARS external validation cohort |
|---|---|---|---|---|
| Potency of initial statin prescribed† | | | | |
| Low | N (%) | 30,310 (23.6) | 12,994 (23.6) | 11,7107 (68.5) |
| Medium | | 90,514 (70.6) | 38,747 (70.5) | 47,825 (28.0) |
| High | | 7,424 (5.8) | 3,224 (5.9) | 5,972 (3.5) |
| Missing data | | | | |
| Missing body mass index | N (%) | 57,782 (45.1) | 24,641 (44.8) | N/A |
| Missing systolic blood pressure | N (%) | 15,076 (11.8) | 6,354 (11.6) | N/A |
| Missing diastolic blood pressure | N (%) | 15,029 (11.7) | 6,337 (11.5) | N/A |

* According to the most recent Hong Kong population census, about 92% of its residents are Han Chinese.

† Statin potency–see **S2 Table** for detailed definitions.

CDARS: Clinical Data Analysis and Reporting System; CPRD: Clinical Practice Research Datalink; HK: Hong Kong; IQR: interquartile range; LDL-C: low-density lipoprotein cholesterol; SD: standard deviation; UK: United Kingdom.

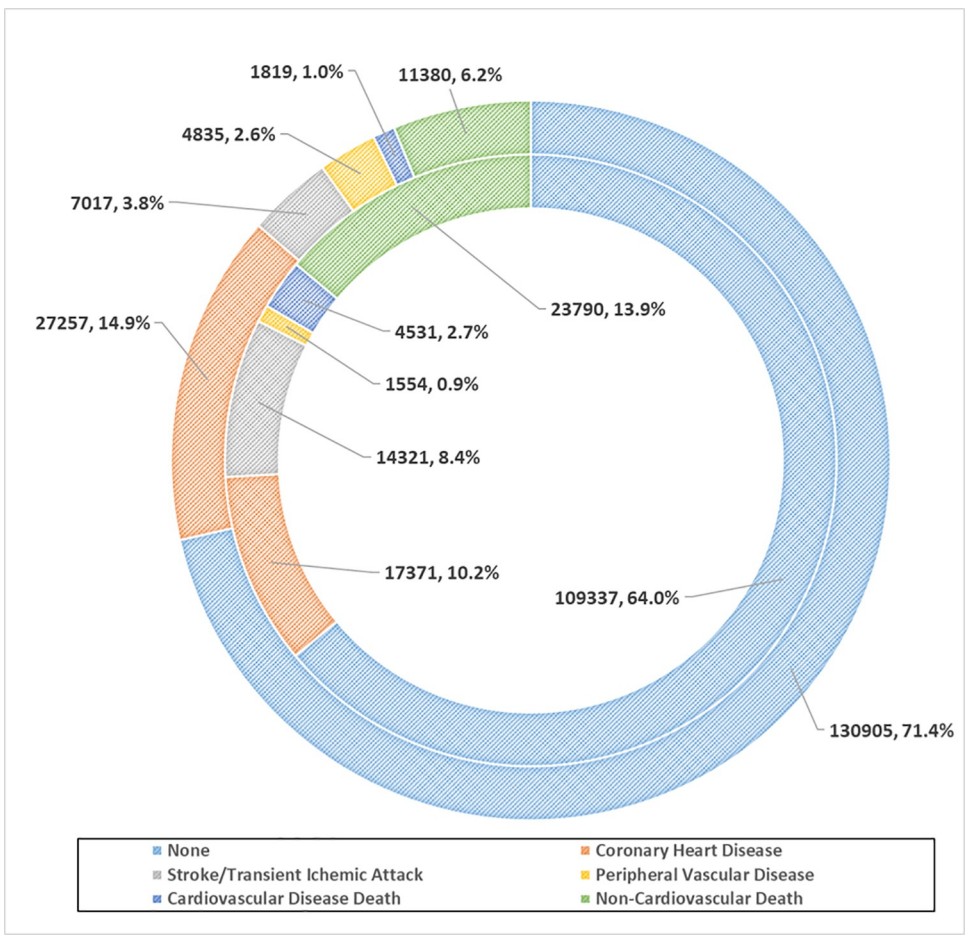

**Fig 2.**

(22.1%) in the HK population. There were some notable differences between the proportion of stroke/TIA and CHD between cohorts, the most common MACE outcomes. There was a higher proportion of CHD (14.9%) and PVD (2.6%) outcomes in UK patients compared to HK patients (10.2% CHD, 0.9% PVD). There was a lower proportion of stroke/TIA (3.8%) in UK patients compared to HK patients (8.4%). The proportion of non-CVD death in the HK patients (13.9%) was twice that of UK patients (6.2%).

## Development of sub-optimal LDL-C statin response model

The multivariable logistic model using the derivation cohort retained age, sex, atrial fibrillation (AF), diabetes, dyslipidaemia, potency or intensity of initial statin prescribed, treated hypertension, prescription of corticosteroid and other lipid lowering medications, LDL-C level at time of statin prescription and medication count. **S3 Table** provides the adjusted odds ratios, with regression coefficients and constants presented in **S4 Table**.

The results of the model showed those with higher baseline levels of LDL-C, atrial fibrillation (in men only), diabetes, treated hypertension, increasing number of concurrent medications (in men only), high potency statins, and other lipid lowering drugs (in men only) were less likely to have sub-optimal LDL-C reduction. Dyslipidaemias and corticosteroids were associated with a higher likelihood of sub-optimal LDL-C reduction. The interaction terms showed that when considered in relation to higher LDL-C levels, those on higher potency statins and had a diagnosis of dyslipidaemias were less likely to have sub-optimal response.

## Validation of sub-optimal LDL-C statin response model

The discriminatory accuracy of the model in predicting sub-optimal LDL-C response performed similarly in both UK cohorts (derivation AUROC 0.704, 95% CI 0.701–0.707, internal validation AUROC 0.703 (95% CI 0.698–0.707) with a slightly lower performance in the HK cohort (external validation AUC 0.677 (95% CI 0.675–0.680) (**Table 2**).

Calibrations plots (**S1 Fig**) show that the model was perfectly calibrated (slope = 1.0) for UK populations by comparing the predicted and observed probability of sub-optimal statin response across each tenth of predicted probabilities. For HK populations, the model under-predicted risk with predicted probabilities consistently lower than observed probabilities (slope = 0.86). An adjustment to the intercept term of the risk model in the HK population would improve its calibration.

The sensitivity and specificity plots for the model are presented in **S2 Fig**. The optimum trade-off occurred at 50% probability as determined by the maximum product index [20] between sensitivity and specificity (shown at the intersection), and hence we classified individuals with SR propensity >50% as "predicted sub-optimal response" and individuals ≤50% as "predicted optimal response". This binary classification resulted in a sensitivity of 64% (95% CI 63.4% - 64.5%) and specificity of 64.9% (95% CI 64.3% - 65.5%).

**Table 2. Model discrimination for sub-optimal LDL-C response to statins.**

| Statin Optimisation Model | Sample Size | AUROC (95% CI) | Standard Error* |
|---|---|---|---|
| Derivation cohort: UK CPRD | 128,248 | 0.704 (0.701–0.707) | 0.002 |
| Internal validation cohort: UK CPRD | 54,965 | 0.703 (0.698–0.707) | 0.002 |
| External validation cohort: HK CDARS | 170,904 | 0.677 (0.675–0.680) | 0.001 |

*Jack-knife procedure to estimate standard errors.

AUROC (c-statistic): area under the receiver operating curve; CDARS: Clinical Data Analysis and Reporting System; CI: confidence interval; CPRD: Clinical Practice Research Datalink; HK: Hong Kong; UK: United Kingdom.

**Table 3. Model performance stratified by statin adherence rates over 2-year follow-up in the United Kingdom CPRD validation cohort (n = 54,965).**

| Statin Adherence Rate (%)* | Mean reduction in LDL-C (SD), mg/dL | Percentage mean reduction in LDL-C (%) | Percentage achieving targeted 40% reduction in LDL-C | AUROC c-statistic (95% CI) |
|---|---|---|---|---|
| <20% | -17.1 (36.7) | -19.1% | 9.9% | 0.637 (0.606–0.670) |
| 20-< 40% | -30.6 (41.6) | -35.8% | 22.2% | 0.633 (0.613–0.653) |
| 40-< 60% | -44.2 (61.3) | -53.7% | 35.0% | 0.642 (0.627–0.658) |
| 60-< 80% | -54.6 (40.5) | -69.9% | 48.2% | 0.672 (0.661–0.684) |
| ≥80% | -65.7 (76.7) | -81.6% | 58.2% | 0.737 (0.731–0.742) |

* Proxy marker for adherence by medication possession count over two-year follow-up from each patient's prescribing record. Estimation of start/stop dates representing discontinuation periods are based on statin pack size and prescribed daily dosage.

AUROC: area under the receiver operating curve; CI: confidence interval; CPRD: Clinical Practice Research Datalink; LDL-C: low-density lipoprotein cholesterol; SD: standard deviation.

## Impact of adherence on model

Approximately 60% of the UK validation cohort (n = 33,491) had a discontinuation period over two-years from starting statins. The statin adherence rate had explained 11% of the variation in the mean reduction in LDL-C (Pearson's r = -0.33; $R^2$ = 0.108) and followed an expected inverse association with both absolute and percentage mean reduction in LDL-C and positive association with percentage patients achieving the targeted 40% reduction in LDL-C from baseline (**Table 3**). The model performance ranged from AUROC of 0.637 (95% CI 0.606–0.670) for patients who had <20% adherence to an AUROC of 0.737 (95% CI 0.731–0.742) in patients who had ≥80% adherence.

## Stratifying groups by LDL-C statin response and CVD risk

Ten-year CVD risk recommended by ESC guidelines and two-year sub-optimal LDL-C reduction propensity was calculated for all patients in both cohorts from baseline risk factors (histograms and distributions shown in **S3 Fig**). By jointly stratifying patients into "high" (> 5%) or "low" (≤ 5%) CVD risk and "sub-optimal" (> 50%) or "optimal" (≤ 50%) LDL-C reduction, four groups for statin response (SR) were created (shown in **S4 Fig**).

For the UK cohort, 45,684 patients (24.9%) were predicted to have optimal statin response and low CVD risk (SR1), 64,997 patients (35.5%) were predicted to have sub-optimal statin response and low CVD risk (SR2), 48,460 patients (26.5%) were predicted to have optimal statin response and high CVD risk (SR3), and 24,072 patients (13.1%) were predicted to have sub-optimal statin response and high CVD risk. For the HK cohort, 47,842 patients (27.9%) were predicted to have optimal statin response and low CVD risk (SR1), 61,725 patients (36.1%) were predicted to have sub-optimal statin response and low CVD risk (SR2), 43,514 patients (25.4%) were predicted to have optimal statin response and high CVD risk (SR3), and 17,823 patients (10.4%) were predicted to have sub-optimal statin response and high CVD risk.

Characteristics of the groups for the UK patients are provided in **S5 Table** and HK patients in **S6 Table**. For UK patients, classified as being at low CVD risk, patients predicted to have sub-optimal compared to optimal response were more likely to be male (55.6% compared to 41.5%), less likely to be treated for hypertension (17.5% compared to 26.7%), with lower mean LDL-C (133.4 mg/dL compared to 187.6 mg/dL), and more frequently initiated on a low potency statin (34.2% compared to 5.4%). Following similar trends, UK patients classified as being at high CVD risk, patients predicted to have sub-optimal response were more likely to be male (64.6% compared to 52.6%), with lower mean LDL-C (123.2 mg/dL compared to

174.9 mg/dL), less likely to be treated for hypertension (27.4% compared to 35.4%), and were much more likely to initiated on a low potency statin (52.8% compared to 12.2%).

In HK patients, SR groups were also significantly different across groups but with some differences compared to UK patients. In HK patients classified as being at low CVD risk, patients predicted to have sub-optimal compared to optimal response were more likely to be male (53.3% compared to 32.1%), lower mean LDL-C (207.4 mg/dL compared to 259.8 mg/dL), less likely to be diabetic (14.2% compared to 23.4%), and less likely to be treated for hypertension (7.1% compared to 14.5%). Similarly, HK patients predicted to have sub-optimal compared to optimal response in high CVD risk groups also were more likely to be male (70.9% compared to 47.6%), have lower mean LDL-C (113.6 mg/dL compared to 166.7 mg/dL), less likely to be diabetic (11.0% compare to 24.2%) or treated for hypertension (10.1% compared to 22.0%). Potency of statin prescribing remained stable in HK patients across all groups, favouring lower dosage statins compared to UK patients. Similar to UK patients, HK patients in high CVD risk groups were generally older, with higher numbers of men, medication count, and smokers.

## Two-year follow-up of LDL-C reduction

Baseline and 2-year follow-up LDL-C levels after statin initiation for UK and HK patients are shown in **Fig 3**. Higher baseline LDL-C levels resulted in more prominent reductions in LDL-C over two years. In UK patients, low and high CVD risk groups, respectively, achieved 40.5% and 41.9% reduction in mean LDL-C in predicted optimal response groups whereas the mean reduction in LDL-C in predicted sub-optimal responders was only 26.8% and 28.6%. In HK patients, low and high CVD risk groups, respectively, achieved 35.0% and 36.7% reduction in mean LDL-C in predicted optimal response groups whereas the mean reduction in LDL-C in predicted sub-optimal responders was only 20.5% and 22.4%. Overall, variability of the LDL-C response in patients in both UK and HK cohorts remained quite high ranging from 60% reduction in LDL-C to increase of 50% in LDL-C after two years of treatment (**S5 Fig**).

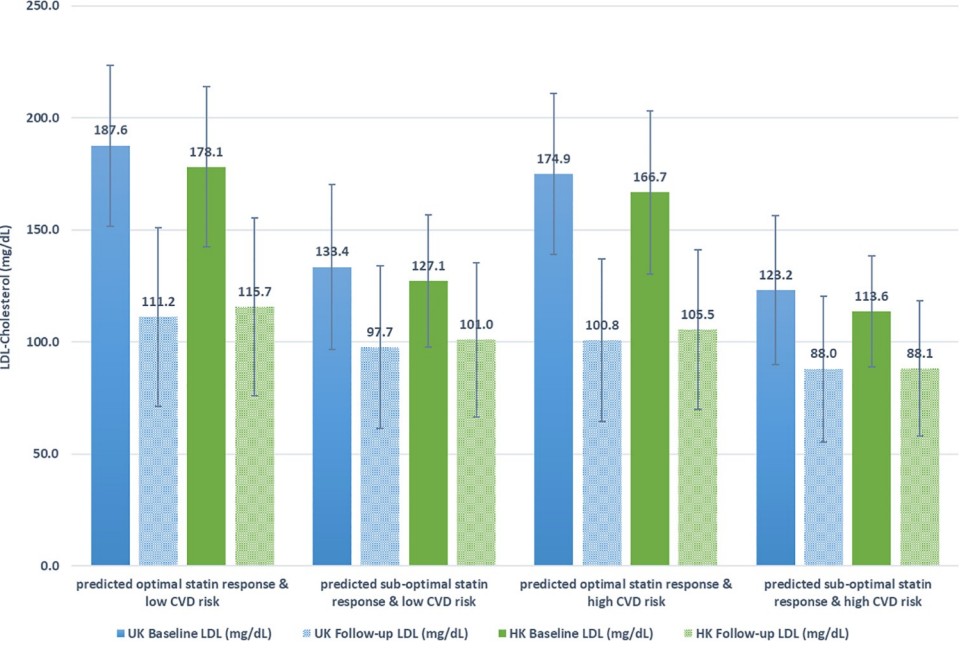

**Fig 3.**

### Long-term follow-up MACE outcomes

Incidence rates (per 1000 person-years) over 10-years for MACE increased significantly across groups (S7 Table). Increasing incidence across SR groups followed similar trends for all-cause mortality, CHD, stroke/TIA, PVD and CVD death for both cohorts. CHD was the most common MACE event to occur during follow-up. Cumulative incidence graphs for 10-year MACE are displayed in Fig 4. In high CVD risk groups, the absolute difference in MACE cumulative event rate at the end of 10-year follow-up was 8% greater in UK patients and 11% greater in HK patients between predicted sub-optimal statin responders (SR4) and predicted optimal statin responders (SR3). Even in low-risk CVD groups, the absolute difference in MACE cumulative event rate at the end of 10-year follow-up was 5% greater in UK patients and 7% greater in HK patients between predicted sub-optimal statin responders (SR2) and predicted optimal statin responders (SR1).

Ten-year models (Table 4) showed that predicted sub-optimal SR in patients with low CVD risk (comparing SR2 to SR1), resulted in hazard ratios for MACE of 1.39 (95% CI 1.35–1.43, p < 0.001) for UK patients and 1.14 (95% CI 1.11–1.17, p < 0.001) for HK patients. Predicted sub-optimal SR in patients with high CVD risk (comparing SR4 to SR3) resulted in hazard ratios for MACE of 1.36 (95% CI 1.32–1.40, p < 0.001) for UK patients and 1.25 (95% CI 1.21–1.28, p < 0.001) for HK patients. In UK patients, the risk of constituent MACE outcomes increased significantly in patients predicted to be sub-optimal responders for CHD, stroke/TIA, and PVD in both high and low CVD risk groups. There was an increased risk of all-cause mortality in UK patients in high CVD risk groups but not in low CVD risk groups. There was no significant difference in CVD death in UK patients. In HK patients, risk of CHD was increased in both low and high CVD risk groups in predicted sub-optimal SR groups. Stroke/TIA and PVD was, however, only increased for sub-optimal SR in high CVD risk groups. Sub-optimal SR groups had significantly lower risks of CVD death and all-cause mortality in patients with low CVD risk (HR<1.0), which can be explained by this group being younger and healthier.

## Discussion

This international study in two countries has identified clinical characteristics of patients that predict variation in LDL-C in response to statin therapy commenced for primary prevention of CVD. We have found that irrespective of their baseline CVD risk, patients with poorer LDL-C response experience much greater rates and risk of adverse cardiovascular outcomes than those likely to achieve target reductions in LDL-C. Groups who have low predicted likelihood to achieve LDL-C treatment targets will experience overall 5–10 more CVD events per 1000 person-years with low baseline CVD risk. The effect is even greater in populations with high baseline CVD risk, with 19–20 more CVD events per 1000-years, compared to population groups with predicted optimal response.

An individual's response to statins should be considered alongside cardiovascular risk assessment to enhance cholesterol treatment strategies and to inform shared decision-making with patients as part of primary prevention of CVD. To our knowledge, this is the first study to demonstrate sub-optimal LDL-C response to statins can be reliably predicted. Our statin response model uses patient characteristics that are routinely available in electronic health records and was robust in both validation cohorts. Considering the effects of adherence, we showed the model predicts LDL-C response when patients are more compliant to their medicines, suggesting there is biological variation in treatment response. In less compliant individuals, expectedly, there is less certainty in predicting LDL-C response.

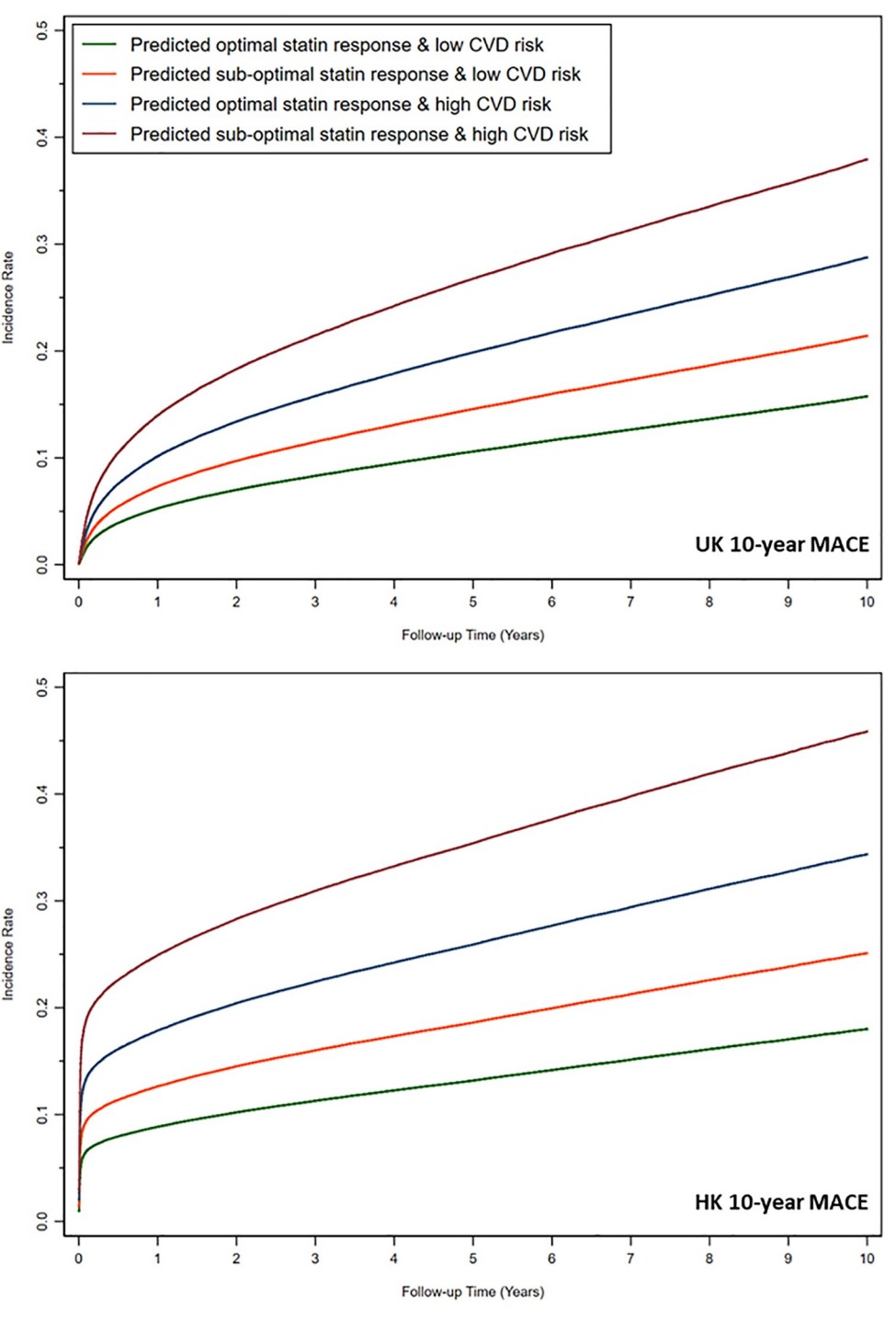

**Fig 4.**

Statin response was associated with age, dyslipidaemias, potency of prescribed statin, base-line LDL-C level and the use of other medications (other lipid therapies and corticosteroids). These factors have been previously found to be associated with variability in LDL-C response [21, 22]. Corticosteroids are usually prescribed to patients with conditions associated with higher levels of inflammation and thus interact with the effectiveness of statins. Patients in

**Table 4. Hazard ratios (95% confidence intervals) for 10-year major adverse cardiovascular events and all-cause mortality for predicted statin response and cardiovascular risk groups.**

| Group | MACE | All-Cause Mortality | Specific CVD Outcomes | | | |
|---|---|---|---|---|---|---|
| | | | CHD | Stroke/TIA | PVD | CVD Death |
| **United Kingdom CPRD cohort 10-year outcomes** | | | | | | |
| *Number of events, N (%)* | 40,928 (22.3) | 13,199 (7.2) | 27,257 (14.9) | 7,017 (3.8) | 4,835 (2.6) | 1,819 (1.0) |
| SR2: Predicted sub-optimal statin response & low CVD risk† | 1.39 (1.35–1.43) * | 0.94 (0.88–1.01) | 1.45 (1.40–1.50) * | 1.30 (1.21–1.41) * | 1.27 (1.16–1.38) * | 0.86 (0.72–1.03) |
| SR4: Predicted sub-optimal statin response & high CVD risk‡ | 1.36 (1.32–1.40) * | 1.10 (1.06–1.15) * | 1.47 (1.42–1.52) * | 1.24 (1.16–1.32) * | 1.18 (1.09–1.28) * | 1.10 (0.98–1.23) |
| **Hong Kong CDARS cohort 10-year outcomes** | | | | | | |
| *Number of events, N (%)* | 37,777 (22.1) | 28,321 (16.5) | 17,371 (10.2) | 14,321 (8.4) | 1,554 (0.9) | 4,531 (2.7) |
| SR2: Predicted sub-optimal statin response & low CVD risk† | 1.14 (1.11–1.17) * | 0.78 (0.75–0.82) * | 1.51 (1.45–1.57) * | 1.00 (0.95–1.04) | 0.95 (0.83–1.09) | 0.73 (0.66–0.81) * |
| SR4: Predicted sub-optimal statin response & high CVD risk‡ | 1.25 (1.21–1.28) * | 1.06 (1.03–1.09) * | 1.69 (1.60–1.78) * | 1.10 (1.04–1.16) * | 1.17 (1.00–1.37) * | 0.97 (0.90–1.05) |

*p <0.05.

†reference group: SR1-predicted optimal statin response & low CVD risk.

‡reference group: SR3-predicted optimal statin response & high CVD risk.

CDARS: Clinical Data Analysis and Reporting System; CHD: coronary heart disease; CPRD: Clinical Practice Research Datalink; CVD: cardiovascular disease; MACE: major adverse cardiovascular event; PVD: peripheral vascular disease; TIA: transient ischemic attack.

both cohorts who were less likely to achieve targeted LDL-C reduction were also more likely to have lower baseline LDL-C levels and be initiated on low potency statins. However, there was greater use of more potent statins in the UK cohort compared to the HK cohort, while greater use of other non-statin lipid lowering drugs in the HK cohort compared to the UK cohort.

Recent updates to cholesterol guidelines have broadened the population of individuals eligible for statins for primary prevention [3, 10, 11, 17]. For example, the UK NICE guideline [3] now suggest statin therapy be considered for individuals who have a cardiovascular risk score > 10%, which has been reduced from a previous threshold of > 20%. In the current study, over half of patients who initiated statins had a sub-optimal LDL-C response in both HK and UK. This would equate to about 6.1 million of the estimated 12 million adults aged 30–84 in the UK who could be eligible for statins, for primary prevention [23].

Other factors, such as physical activity, diet and statin or treatment side-effects (which are less reliably recorded) should clearly also be considerations alongside statin alternatives to achieve shared clinical decision making with patients for appropriate cholesterol management. However, identifying patients likely to have sub-optimal LDL-C response to statins could enable clinicians to be more aggressive in cholesterol management by identifying those who need closer monitoring, or initiated on higher statin doses, given in combination with ezetimibe [24]. Recent evidence suggests that reduction in cardiovascular events is significant even to ultra-low levels of LDL-C less than 0.2 mmol/L (7.7 mg/dL) [25]. Our study shows that many patients are initiated on low dose statins in the predicted sub-optimal groups suggesting many clinicians may not be aware of the clinical benefits of a stronger treatment regime even in those with lower baseline levels of LDL-C with high risk of CVD.

Medications that aim to reduce LDL-C such as PCSK9 inhibitors and Ezetimibe [26, 27] are presently indicated for both primary and secondary prevention of CVD. When taken in combination with statin and diet therapies, additional reduction in LDL-C levels have been shown in trials.

The strengths of this study include its large sample size, long-term follow-up, from general patient populations in two countries, derived from high quality electronic patient health

records. Data linkages to hospital records and death registries enhanced the quality of outcome ascertainment for the cohorts of patients derived from these databases. Although the model was developed using a UK cohort, the applicability of the model in the ethnically different HK cohort, enhances external validity and generalisability of more aggressive lipid management across both Western and Asian populations. By adopting an open cohort design and accounting for statin type and intensity, the predicted statin response and event rates across time periods reflect real-world changes in practice over time. Both populations have experienced prescribing trends towards stronger statin intensity over time [7, 28].

Some limitations of this study are recognized. Although the proxy for statin adherence explained a third of the variation in LDL-C response, prescribing records cannot confirm whether a patient actually takes their medication, which is a shared limitation of all observational studies and trials. Moreover, patients who are on cardioprotective therapies are likely to be on other medications other than statins and are more likely to discontinue multiple therapies, which has also been shown to influence the mortality outcomes [29]. In common with other large population studies using routinely collected data, this limitation together with others including ascertainment and information bias, potential bias due to missing data and unmeasured confounding is acknowledged. Newer medications such as PCSK9 are not yet indicated for primary prevention, hence no data was available to assess its effect on cholesterol response. We used the ESC guidelines to stratify CVD risk due to its simplicity, accessibility to the published risk equations, and being recommended by ESC and HK guideline committees as acceptable for use in both the UK and HK. We acknowledge that different national guidelines may recommend use of other CVD risk equations. Finally, we did not consider the effects of medication switching during follow-up. Future analysis could consider modelling the effects of time-varying exposure periods to account for any changes of dosing.

## Conclusions

Predicted sub-optimal LDL-C lowering response to statins consistently resulted in higher rates of MACE. To enhance cholesterol management for primary prevention of CVD, clinicians should consider a patient's likely response to statins as a component of CVD risk assessment.

## Supporting information

**S1 Fig.** Validation plots for logistic model applied in (a) UK CPRD internal validation cohort, n = 54,985 and (b) HK CDARS external validation cohort, n = 170,904. E:O–log of the expected/observed number of events; CITL–calibration-in-the-large; AUC–area under the curve; slope–calibration slope. The circles represent deciles of patients grouped by similar predicted risk. The distribution of patients (stratified by outcome) is indicated with spikes at the bottom of the graph. Patients with sub-optimal response are represented by spikes above the x-axis (red line), and patients with optimal response, below the x-axis).
(DOCX)

**S2 Fig. Plot of sensitivity and specificity to determine optimum cut-off for the classification of sub-optimal LDL response risk to statins using the standard approach model in the UK CPRD validation cohort (n = 54,985).**
(DOCX)

**S3 Fig. Histograms of predicted 10-year CVD risk score by ESC guidelines and 2-year propensity of sub-optimal LDL-C reduction for UK and HK cohorts.**
(DOCX)

**S4 Fig. Joint stratification of 10-year CVD risk by ESC guidelines and 2-year propensity of sub-optimal LDL-C reduction in both UK and HK patient cohorts.** ESC: European Society of Cardiology CVD SCORE >5% is defined as high CVD risk; Low-density lipoprotein cholesterol statin response propensity of > 50% is defined as sub-optimal response.
(DOCX)

**S5 Fig. Waterfall plots of 2-year percentage LDL-C reduction from baseline based on initiated statin potency.** Potency classification by expected LDL-C reduction (see **S2 Table** for specific statin types); 20%–30%: low potency; 31%–40%: medium potency; Above 40%: high potency.
(DOCX)

**S1 Table. Potential predictors of sub-optimal statin response propensity model.** The choice of medications of medications is informed by review of clinical guidelines and previous literature on CVD risk (UK NICE Guidelines, QRISK3).
(DOCX)

**S2 Table. Grouping of statins by potency.** [1] 20%–30%: low intensity; [2] 31%–40%: medium intensity; [3] Above 40%: high intensity. Note: combination therapies with a statin and ezetimibe 10 mg would push potency up one group (i.e., low to medium, medium to high).
(DOCX)

**S3 Table. Multivariable logistic regression for mutually adjusted diagnostic variables of sub-optimal LDL-C response to statins using UK CPRD derivation cohort (n = 128,298).** $p < 0.05$; No–reference category for dichotomous variables.
(DOCX)

**S4 Table. Logistic regression coefficients and constants derived from the UK Clinical Practice Research Datalink (CPRD) for determining sub-optimal LDL-C response to statistics.** Unit: per mmol/L.
(DOCX)

**S5 Table. Characteristics of 183,213 patients from the UK CPRD dataset by statin response group based on ESC score.** + Significance determined by Kruskal-Wallis non-parametric H test between two or more groups; ESC: European Society of Cardiology; CPRD: Clinical Practice Research Datalink; HDL: high-density lipoprotein; LDL: low-density lipoprotein; CVD: cardiovascular disease; SR1 –Patients with predicted optimal statin response and low CVD risk; SR2 –Patients with predicted sub-optimal statin response and low CVD risk; SR3 – Patients with predicted optimal statin response and high CVD risk; SR4 –Patients with predicted sub-optimal statin response and high CVD risk.
(DOCX)

**S6 Table. Characteristics of 170,904 patients from the HK CDARS dataset by statin response group based on ESC score.** + Significance determined by Kruskal-Wallis non-parametric H test between two or more groups; ESC: European Society of Cardiology; CPRD: Clinical Practice Research Datalink; HDL: high-density lipoprotein; LDL: low-density lipoprotein; CVD: cardiovascular disease; SR1 –Patients with predicted optimal statin response and low CVD risk; SR2 –Patients with predicted sub-optimal statin response and low CVD risk; SR3 –Patients with predicted optimal statin response and high CVD risk; SR4 –Patients with predicted sub-optimal statin response and high CVD risk.
(DOCX)

**S7 Table. 10-year incidence rates (per 1000 person-years) for major adverse cardiovascular events (MACE) and all-cause mortality for predicted statin response and ESC cardiovascular risk groups.** 95% Confidence Intervals are provided in () for all incidence rates. SR1 – Patients with predicted optimal statin response and low CVD risk; SR2 –Patients with predicted sub-optimal statin response and low CVD risk; SR3 –Patients with predicted optimal statin response and high CVD risk; SR4 –Patients with predicted sub-optimal statin response and high CVD risk. ESC: European Society of Cardiology; CPRD: Clinical Practice Research Datalink; CDARS: Clinical Data Analysis and Reporting System; CVD: cardiovascular disease; TIA: Transient Ischemic Attack; CHD: Coronary Heart Disease; PVD: Peripheral Vascular Disease; UK: United Kingdom; HK: Hong Kong.
(DOCX)

**S1 Text. Supplemental methods.**
(DOCX)

## Acknowledgments

We are grateful to all the patients contributing data to both CPRD and CDARS.

## Author Contributions

**Conceptualization:** Stephen Franklin Weng, Nadeem Qureshi, Joe Kai.

**Data curation:** Stephen Franklin Weng, Ralph Kwame Akyea, Kenneth KC Man, Wallis C. Y. Lau, Joseph Edgar Blais.

**Formal analysis:** Stephen Franklin Weng, Ralph Kwame Akyea, Kenneth KC Man, Wallis C. Y. Lau.

**Funding acquisition:** Stephen Franklin Weng.

**Investigation:** Stephen Franklin Weng, Kenneth KC Man, Wallis C. Y. Lau, Barbara Iyen.

**Methodology:** Stephen Franklin Weng, Ralph Kwame Akyea, Kenneth KC Man, Wallis C. Y. Lau, Nadeem Qureshi, Ian C. K. Wong, Joe Kai.

**Project administration:** Stephen Franklin Weng.

**Resources:** Stephen Franklin Weng.

**Supervision:** Stephen Franklin Weng, Ian C. K. Wong.

**Validation:** Stephen Franklin Weng.

**Visualization:** Stephen Franklin Weng, Ralph Kwame Akyea.

**Writing – original draft:** Stephen Franklin Weng, Ralph Kwame Akyea.

**Writing – review & editing:** Stephen Franklin Weng, Ralph Kwame Akyea, Kenneth KC Man, Wallis C. Y. Lau, Barbara Iyen, Joseph Edgar Blais, Esther W. Chan, Chung Wah Siu, Nadeem Qureshi, Ian C. K. Wong, Joe Kai.

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
