## [Decision Letter · Decision Letter 0]

6 Jul 2021

PONE-D-21-11502

Determining propensity for sub-optimal low-density lipoprotein cholesterol response to statins and future risk of cardiovascular disease

PLOS ONE

Dear Dr. Akyea,

Thank you for submitting your manuscript to PLOS ONE. After careful consideration, we feel that it has merit but does not fully meet PLOS ONE’s publication criteria as it currently stands. Therefore, we invite you to submit a revised version of the manuscript that addresses the points raised during the review process.

We look forward to receiving your revised manuscript.

Kind regards,

Carmine Pizzi

Academic Editor

PLOS ONE

"JB is supported by the Hong Kong Research Grants Council as a recipient of the Hong Kong

PhD Fellowship Scheme."

3. Thank you for stating the following in the Competing Interests/Financial Disclosure * (delete as necessary) section:

"Funded by scientific donation by AMGEN Ltd. The funders of the study had no role in study design, data collection, data analysis, data interpretation, or writing of the report."

We note that you received funding from a commercial source: AMGEN Ltd.

"NQ is a member of the most recent NICE Familial Hypercholesterolaemia & Lipid Modification Guideline Development Groups (CG71 & CG181). SW, IW are members of the Clinical Practice Research Datalink (CPRD) Independent Scientific Advisory Committee (ISAC). RKA currently holds an NIHR-SPCR funded studentship (2018-2021). SW reports honorarium from AMGEN and is also an employee of Janssen. KM holds the CW Malpethorpe Fellowship Award and reports personal fees from IQVIA Ltd have been received outside the submitted work. EC and IW report grants from Amgen. The remaining authors have no competing interests."

Review Comments to the Author

Reviewer #1: I’ve read with attention the paper of Ralph Kwame Akyea et al. that is potentially of interest. The background and aim of the study have been clearly defined. The methodology applied is overall correct and the results are reliable. I’ve only some minor comments: - The difference in use of low intensity statin in the two cohort could be influenced by the fact that Asian are more sensible to statin-side effects and some drugs are to be used at lower dosages in Asians

- The authors stress that ezetimibe should be used in secundary prevention patients only... why? The guidelines suggest to add ezetimibe where LDL-C target is not achieved, in high-risk patients in primary prevention as well

- The authors suggest the use of icosapent ethyl as an adjunct to diet in adults with triglycerides of at least 500 mg/dL. This is not supported by the reference chosen by the authors nor it is supported by any guideline. Probably fenofibrate should be considered as well.

---

## [Author Response · Author response to Decision Letter 0]

29 Jul 2021

Response to Comments from Editorial Committee

Comment:

Response:

The manuscript has been formatted/revised to meet PLOS ONE’s style requirements, including those for file naming.

Comment:

"JB is supported by the Hong Kong Research Grants Council as a recipient of the Hong Kong

PhD Fellowship Scheme."

Response:

The funding statement "JB is supported by the Hong Kong Research Grants Council as a recipient of the Hong Kong PhD Fellowship Scheme" has been removed from the acknowledgement section to the competing interests/financial disclosure section.

The funding statement remains the same: “Funded by scientific donation by AMGEN Ltd. The funders of the study had no role in study design, data collection, data analysis, data interpretation, or writing of the report”. 

Comment:

3. Thank you for stating the following in the Competing Interests/Financial Disclosure * (delete as necessary) section:

"Funded by scientific donation by AMGEN Ltd. The funders of the study had no role in study design, data collection, data analysis, data interpretation, or writing of the report."

We note that you received funding from a commercial source: AMGEN Ltd.

Response:

The competing interests/financial disclosure section has been revised accordingly and now includes the statement "This does not alter our adherence to PLOS ONE policies on sharing data and materials.”, as required. The competing interest statement now reads (page 22): 

“NQ is a member of the most recent NICE Familial Hypercholesterolaemia & Lipid Modification Guideline Development Groups (CG71 & CG181). SW, IW are members of the Clinical Practice Research Datalink (CPRD) Independent Scientific Advisory Committee (ISAC). RKA currently holds an NIHR-SPCR funded studentship (2018-2021). SW reports honorarium from AMGEN and is also an employee of Janssen. KM holds the CW Malpethorpe Fellowship Award and reports personal fees from IQVIA Ltd have been received outside the submitted work. JB is supported by the Hong Kong Research Grants Council as a recipient of the Hong Kong PhD Fellowship Scheme. EC and IW report grants from Amgen. The remaining authors have no competing interests. This does not alter our adherence to PLOS ONE policies on sharing data and materials.”

Comment:

"NQ is a member of the most recent NICE Familial Hypercholesterolaemia & Lipid Modification Guideline Development Groups (CG71 & CG181). SW, IW are members of the Clinical Practice Research Datalink (CPRD) Independent Scientific Advisory Committee (ISAC). RKA currently holds an NIHR-SPCR funded studentship (2018-2021). SW reports honorarium from AMGEN and is also an employee of Janssen. KM holds the CW Malpethorpe Fellowship Award and reports personal fees from IQVIA Ltd have been received outside the submitted work. EC and IW report grants from Amgen. The remaining authors have no competing interests."

Response:

As indicated in the response to the earlier comment, the competing interest section has been revised as required and the sentence “The remaining authors have no competing interests. This does not alter our adherence to PLOS ONE policies on sharing data and materials.” has been added (page 20).

Comment:

Response:

There are ethical and legal restrictions on sharing de-identified data set and this has been explained in the revised data availability section (page 22):

“The third-party data for this study were obtained from the Clinical Practice Research Datalink (CPRD) and the University of Hong Kong/Hospital Authority Hong Kong West Cluster. CPRD is a research service that provides primary care and linked data for public health research. CPRD and University of Hong Kong/Hospital Authority data governance and our own license to use data do not allow us to distribute or make available patient data directly to other parties. However, data is available upon application to CPRD (www.cprd.com) and the University of Hong Kong/Hospital Authority (www.ha.org.hk).” 

Response to Comments from Reviewer 1

Specific comments

Comment:

“I’ve read with attention the paper of Ralph Kwame Akyea et al. that is potentially of interest. The background and aim of the study have been clearly defined. The methodology applied is overall correct and the results are reliable. I’ve only some minor comments: - The difference in use of low intensity statin in the two cohort could be influenced by the fact that Asian are more sensible to statin-side effects and some drugs are to be used at lower dosages in Asians”

Response:

We thank the reviewer for the positive assessment.

Comment:

“The authors stress that ezetimibe should be used in secondary prevention patients only... why? The guidelines suggest adding ezetimibe where LDL-C target is not achieved, in high-risk patients in primary prevention as well”

Response:

Thank you for highlighting this. We agree with the reviewer that the guidelines recommend Ezetimibe for primary prevention as well. The sentence has, therefore, been revised to reflect this. The sentence now reads (page 14): “Medications that aim to reduce LDL-C such as PCSK9 inhibitors and Ezetimibe, [26,27] are presently indicated for both primary and secondary prevention of CVD.”

Comment:

“The authors suggest the use of icosapent ethyl as an adjunct to diet in adults with triglycerides of at least 500 mg/dL. This is not supported by the reference chosen by the authors, nor it is supported by any guideline. Probably fenofibrate should be considered as well.”

Response:

The paragraph have been revised and the bit about the use of icosapent ethyl as an adjunct to diet in adults with triglycerides of at least 500mg/dL, has been removed. The paragraph now reads (page 14): “Medications that aim to reduce LDL-C such as PCSK9 inhibitors and Ezetimibe, [26,27] are presently indicated for both primary and secondary prevention of CVD. When taken in combination with statin and diet therapies, additional reduction in LDL-C levels have been shown in trials.”

---

## [Decision Letter · Decision Letter 1]

18 Nov 2021

Determining propensity for sub-optimal low-density lipoprotein cholesterol response to statins and future risk of cardiovascular disease

PONE-D-21-11502R1

Dear Dr. Akyea,

We’re pleased to inform you that your manuscript has been judged scientifically suitable for publication and will be formally accepted for publication once it meets all outstanding technical requirements.

Kind regards,

Carmine Pizzi

Academic Editor

PLOS ONE

---

## [Editor Report · Acceptance letter]

22 Nov 2021

PONE-D-21-11502R1 

Determining propensity for sub-optimal low-density lipoprotein cholesterol response to statins and future risk of cardiovascular disease 

Dear Dr. Akyea:

I'm pleased to inform you that your manuscript has been deemed suitable for publication in PLOS ONE. Congratulations! Your manuscript is now with our production department. 

Kind regards, 

on behalf of

Prof Carmine Pizzi 

Academic Editor

PLOS ONE